# Highly Sensitive, Selective, Flexible and Scalable Room-Temperature NO_2_ Gas Sensor Based on Hollow SnO_2_/ZnO Nanofibers

**DOI:** 10.3390/molecules26216475

**Published:** 2021-10-27

**Authors:** Jiahui Guo, Weiwei Li, Xuanliang Zhao, Haowen Hu, Min Wang, Yi Luo, Dan Xie, Yingjiu Zhang, Hongwei Zhu

**Affiliations:** 1Key Laboratory of Material Physics, Ministry of Education, School of Physics and Microelectronics, Zhengzhou University, Zhengzhou 450052, China; guojiahuizzu@163.com (J.G.); luoyilogic@163.com (Y.L.); 2State Key Laboratory of New Ceramics and Fine Processing, School of Materials Science and Engineering, Tsinghua University, Beijing 100084, China; zhao-xl15@mails.tsinghua.edu.cn (X.Z.); haowenh6677@126.com (H.H.); wangmin199303@163.com (M.W.); 3Department of Basic Sciences, Air Force Engineering University, Xi’an 710051, China; kgdliweiwei@163.com; 4Beijing National Research Center for Information Science and Technology (BNRist), Institute of Microelectronics, Tsinghua University, Beijing 100084, China

**Keywords:** electrospinning, tin oxide nanofibers, zinc oxide, gas sensor, flexible devices

## Abstract

Semiconducting metal oxides can detect low concentrations of NO_2_ and other toxic gases, which have been widely investigated in the field of gas sensors. However, most studies on the gas sensing properties of these materials are carried out at high temperatures. In this work, Hollow SnO_2_ nanofibers were successfully synthesized by electrospinning and calcination, followed by surface modification using ZnO to improve the sensitivity of the SnO_2_ nanofibers sensor to NO_2_ gas. The gas sensing behavior of SnO_2_/ZnO sensors was then investigated at room temperature (~20 °C). The results showed that SnO_2_/ZnO nanocomposites exhibited high sensitivity and selectivity to 0.5 ppm of NO_2_ gas with a response value of 336%, which was much higher than that of pure SnO_2_ (13%). In addition to the increase in the specific surface area of SnO_2_/ZnO-3 compared with pure SnO_2_, it also had a positive impact on the detection sensitivity. This increase was attributed to the heterojunction effect and the selective NO_2_ physisorption sensing mechanism of SnO_2_/ZnO nanocomposites. In addition, patterned electrodes of silver paste were printed on different flexible substrates, such as paper, polyethylene terephthalate and polydimethylsiloxane using a facile screen-printing process. Silver electrodes were integrated with SnO_2_/ZnO into a flexible wearable sensor array, which could detect 0.1 ppm NO_2_ gas after 10,000 bending cycles. The findings of this study therefore open a general approach for the fabrication of flexible devices for gas detection applications.

## 1. Introduction

Nitrogen dioxide (NO_2_), one of the most hazardous gases, poses a great threat to humans, animals, and plants [1,2,3]. It may induce various illnesses in humans even at very low concentrations. In addition, the excessive emission of NO_2_ gas causes numerous environmental problems, such as surface water acidification and photochemical smog [4,5]. Therefore, the detection of NO_2_ gas is critical for human health and environmental conservation. The interest in semiconducting metal oxides has grown in recent years owing to their good performance in the optical, electronic and gas sensor fields. Semiconducting metal oxides such as SnO_2_, ZnO, In_2_O_3_, TiO_2_, and NiO have the advantages of good chemical stability, excellent sensitivity, low cost, and easy fabrication. Therefore, these semiconductors have been widely investigated and applied in NO_2_ gas detection [6,7,8,9,10,11]. There are various ways of improving the detection efficiency of these materials, such as the application of an electrostatic field [12], doping with other nanomaterials [13], and ultraviolet (UV) illumination [14,15]. Of these techniques, nanomaterials doping is attracting increasing attention as an effective strategy.

SnO_2_ and ZnO, as typical n-type semiconductors, have been widely regarded as effective and practical gas sensing materials over the past decade [16]. However, most studies on the gas sensing properties of SnO_2_ and ZnO have been carried out at high temperature. Accordingly, it is still challenging to fabricate SnO_2_- and ZnO-based NO_2_ gas sensors with excellent sensing performances at room temperature. Therefore, several reasonable approaches have been used with the aim of improving their sensing properties, including doping with metals [17] and carbon materials [18], and surface modification using Pt, Ag particles [19,20]. SnO_2_ and ZnO nanocomposites are deemed to be effective materials for improving gas sensing properties [16,21,22]. Sunghoon et al. fabricated SnO_2_-Core/ZnO-shell nanowires and found a response of about 239% towards 1 ppm of NO_2_ gas [20]. Yang et al. also fabricated ZnO-SnO_2_ heterojunction nanobelts that showed a faster response (1.8 s)/recovery (18 s) speed to triethylamine [22]. Although ZnO-SnO_2_ nanocomposites have been prepared and modified to improve sensing performance, room temperature chemical sensors still face great challenges during application. In this study, a novel SnO_2_/ZnO nanocomposite with excellent sensing performance at room temperature was prepared. Its sensing properties and mechanism at room temperature were then investigated. Moreover, due to the increased development of wearable electronic devices, it is logical to study flexible wearable gas sensors [23,24]. However, flexible gas sensors usually involve transfer of the prepared substrate layers or even the whole device onto special flexible substrates, as well as the attachment of target gases, limiting their practical applications. Therefore, it would be necessary to develop a facile, flexible, and scalable gas sensor preparation method.

Herein, hollow SnO_2_/ZnO nanofibers were synthesized by a facile electrospinning and calcination method. The gas sensing performance and mechanism of gas sensors based on pristine SnO_2_ and SnO_2_/ZnO nanocomposite were also investigated at room temperature. The results revealed that the SnO_2_/ZnO nanofibers coated on titanium/gold interdigital electrodes exhibited high sensitivity and selectivity to NO_2_ gas sensing at ppb level, and at a minimum detection limit of 0.1 ppm during testing. SnO_2_/ZnO sensors exhibited high sensitivity to 0.5 ppm NO_2_, with a response value of 336% and a fast response time of <2 min, all of which relied on both physisorption and chemisorption-based charge transfer. Furthermore, patterned silver paste electrodes were printed on different flexible substrates including paper, polyethylene terephthalate (PET) and polydimethylsiloxane (PDMS) by using screen printing. These were then integrated with SnO_2_/ZnO nanofibers as sensing layers into a novel flexible and wearable gas sensor. After 10,000 bending cycles, the SnO_2_/ZnO flexible gas sensor did not lose its high sensitivity properties for the detection of NO_2_ gas.

## 2. Experimental Section

### 2.1. Materials

N-dimethylformamide (DMF), stannous chloride (SnCl_4_·2H_2_O), and zinc nitrate hexahydrate (Zn(NO_3_)_2_·6H_2_O) were purchased from Alfa Aesar. Poly (vinylpyrrolidone) (PVP, *M*_w_=1,300,000) was obtained from Sigma-Aldrich. Ethanol and silver paste were supplied by Sinopharm Chemical Reagents (Shanghai, China), and Beijing NANOTOP Co. Ltd. (Beijing, China), respectively.

### 2.2. Synthesis of Hollow SnO_2_ and SnO_2_/ZnO Nanofibers

The transparent precursor solution for the synthesis of SnO_2_ nanofibers was prepared by addition of 0.6 g SnCl_4_·2H_2_O into 10 mL ethanol/DMF (volume ratio 1:1) solvent mixture, in which 0.8 g PVP had already been dissolved. In the preparation of SnO_2_/ZnO nanofibers, 0.005 g, 0.01g and 0.03 g Zn(NO_3_)_2_·6H_2_O were respectively added into two bottles of the precursor solution mentioned above, and the corresponding prepared samples labeled SnO_2_/ZnO-1, SnO_2_/ZnO-2, and SnO_2_/ZnO-3, respectively. The precursor solution was magnetically stirred at room temperature for 3 h until a clear and transparent solution was formed. A 2 mL volume of precursor solution was then drawn using a 5 mL syringe with a stainless steel needle for electrospinning. The needle was connected to the positive pole, and the negative pole was connected to an aluminum foil, which was placed 20 cm away from the collector, as shown in Figure 1a. The electrospinning parameters used were as described below: the voltage was 20 kV, the humidity range was 40–50% relative humidity (RH), and the collection time was 2 h. The electrospun fibers were then calcinated at 600 °C for 3 h at a heating rate of 10 °C/min.

### 2.3. Fabrication of Flexible Patterned Electrodes

A screen plate with patterned electrodes was fixed on a commercial screen-printing press. The different substrates (paper, PET, PDMS) were placed about 5 cm below the screen plate. Silver paste was then pressed along the patterns using a squeegee at a constant speed to form a clear and flat flexible electrode traces on the substrate, as shown in Figure 1b. The flexible substrates with silver electrodes were dried in the oven at 120 °C for 0.5 h after fabrication.

### 2.4. Preparation and Test of Gas Sensors

Titanium/gold (Ti/Au) interdigital electrodes, with gaps and finger widths of both 10μm, were fabricated on silicon/silicon dioxide (Si/SiO_2_) substrates by lithography. The flexible electrodes deposited on paper, PET and PDMS by screen printing had a gap and finger widths of 0.1 mm. Sensing nanofibers were then dispersed in deionized (DI) water, followed by drop-coating onto the electrodes (Figure 1b). The gas sensing properties were evaluated using a homemade system at room temperature (~20 °C). This system could monitor the changes in resistance in dynamic variation process of gas concentration controlled by mass flow controller. The resistance values were recorded using Keithley 2700, China. The response values were then calculated using Equation (1):*S* (%) = (*R*_a_ − *R*_g_)/*R*_a_ × 100%(1)
where *R*_a_ is the initial resistance of air and *R*_g_ is the resistance of the target gas. The response and recovery time were 90% of the time when the resistance reached its maximum in the target gas and the minimum in the air, respectively.

### 2.5. Characterizations

The hollow SnO_2_ nanofibers, SnO_2_/ZnO composite nanofibers were characterized by scanning electron microscopy (SEM, JSM-7401F, JEOL, Akishima, Japan), transmission electron microscopy (TEM, JEM-2010, JEOL, Tokyo, Japan), electron dispersive spectroscopy (EDS) installed in TEM, X-ray photoelectron spectroscopy (XPS, PHI-5300, Perkin-Elmer, Waltham, MA, USA) using AlKα radiation (h*ν* = 1486.6 eV), and specific surface area analyzer (Autosorb-iQ2-MP, Quantachrome, Shanghai, China). The change in resistance of the silver electrodes during bending cycles was recorded using a stretcher (INSTRON 5943, Instron, Shanghai, China).

## 3. Results and Discussion

### 3.1. Characterizations of SnO_2_, SnO_2_/ZnO Nanofibers

The process of SnO_2_/ZnO composite nanofiber preparation is illustrated in Figure 1a. Humidity is a very important factor in the electrospinning process. Low humidity causes the blockage of electrospinning needle, whereas high humidity makes it difficult to collect samples at the collector. Figure 1b displays the fabrication process of flexible electrodes through screen printing using silver paste on different substrates. The fabrication process was strongly influenced by the physical properties of silver paste, especially the viscosity and the organic solvents used. The appropriate organic solvent enabled silver paste to be cured at room temperature. High viscosity was essential in preventing excessive spreading on the substrate, whereas high-viscosity silver paste was not compatible with other printing techniques, such as gravure [25] and inkjet [26].

SEM images of the electrospun SnO_2_/PVP, SnO_2_/ZnO/PVP-1, SnO_2_/ZnO/PVP-2, and SnO_2_/ZnO/PVP-3 nanofibers are shown in Figure 2a–d. The diameters of these fibers were found to be similar to each other, and their surface morphology appeared to be smooth because of the features of the polymer used. SEM images of hollow SnO_2_, SnO_2_/ZnO-1, SnO_2_/ZnO-2, and SnO_2_/ZnO-3 nanofibers after calcination were as shown in Figure 2e–h. The surface of the nanofibers was clearly concave–convex and porous after calcination treatment, and the nanofibers consisted of nanoparticles. By comparing these images, it was revealed that the morphologies of all the hollow nanofibers were porous, and their diameters exhibited almost no significant difference (*d* = 180 ± 20 nm). Therefore, it can be inferred that the slight difference in diameters between SnO_2_, SnO_2_/ZnO nanofibers may not have an appreciable effect on their sensing performance.

To further verify the structure of SnO_2_/ZnO composite nanofibers, TEM images were taken from the as-prepared SnO_2_/ZnO-3 sample. The TEM images in Figure 3a and its inset show porous structures with a mean diameter of 112.8 nm. As shown in Figure 3b, the interplanar spacing was 0.34 nm for the SnO_2_ (110) plane and 0.27 nm for the ZnO (100) plane, which were in accordance to a previous study [26]. This result illustrated that SnO_2_/ZnO composite nanofibers fabricated consisted of SnO_2_ and ZnO nanoparticles of different sizes (*d* = 10 ± 5 nm), which had a significant effect on the gas sensing properties of the nanofibers. The response performance decreased with the increase of grain size and exited from the optimal grain size range [27]. The grain size of SnO_2_ and ZnO nanoparticles prepared in this study was just in the optimal range. To investigate the chemical composition of SnO_2_/ZnO composite nanofibers, element mapping images were also measured. As shown in Figure 3c, the O (white), Sn (green) and Zn (red) atoms were uniformly distributed throughout the SnO_2_/ZnO nanofibers. Moreover, the percentage of Sn atoms was found to be more than that of O and Zn, with Zn being the least.

The surface chemical composition of SnO_2_/ZnO-2 composite nanofibers was further characterized using XPS. As shown in Figure 4a, the peaks of Sn, Zn, O, and C were all observed, and no other impurities were detected. The two peaks that emerged in Figure 4b, located at 486.4 and 494.8 eV with a separation of 8.4 eV, corresponded to Sn 3d_5/2_ and Sn 3d_3/2_ of Sn^4+^ ions, respectively. The two peaks that appeared at a binding energy of 1044.9 and 1021.9 eV in Figure 4c corresponded to Zn 2p_1/2_ and Zn 2p_3/2_ of Zn^2+^ ions, respectively [28,29]. From Figure 4d, it can be seen that the O 1s spectrum consisted of two different components at 530.27 eV and 531.60 eV, corresponding to the lattice oxygen in ZnO and the lattice oxygen in SnO_2_, respectively. These results further demonstrate that SnO_2_/ZnO composite nanofibers were formed from SnO_2_ and ZnO nanoparticles [30].

The fabrication process of hollow SnO_2_ nanofibers has been previously outlined [31]. During the annealing process, due to the decomposition of PVP template, Sn precursors are rapidly oxidized and redistributed through surface diffusion to form SnO_2_ nanoparticles, which is a component of the hollow nanofibers. Similarly, Sn and Zn precursors will be oxidized to form SnO_2_ and ZnO nanoparticles during the calcination process. The data shown in Table 1 point to the conclusion that moderate doping of ZnO remarkably promotes the Sn precursors oxidation, leading to the formation of more SnO_2_ nanoparticles. Moreover, the surface area of SnO_2_/ZnO composite nanofibers is larger than that of pure SnO_2_ nanofibers, and thus it is prone to providing more active sites for the adsorption and desorption of gas molecules [28]. Additionally, the surface area of SnO_2_/ZnO increases with the concentration of ZnO. Therefore, the SnO_2_/ZnO composite nanofibers were expected to have enhanced gas sensing properties.

### 3.2. Gas Sensing Properties of SnO_2_, SnO_2_/ZnO Composite Nanofibers

The gas sensing properties of SnO_2_, SnO_2_/ZnO-1, SnO_2_/ZnO-2 and SnO_2_/ZnO-3 at different concentrations of NO_2_ gas at room temperature were studied. Figure 5 shows that the resistance increased upon exposure to NO_2_ gas and recovered completely to the initial resistance value upon the removal of NO_2_. This indicates that the SnO_2_ and SnO_2_/ZnO gas sensors had good and stable reversibility. The response values of pure SnO_2_ gas sensor exposed to 0.1, 0.5, 5, 10 and 20 ppm of NO_2_ gas were 8%, 13%, 24%, 34% and 45%, respectively. Meanwhile, the response values of the SnO_2_/ZnO-1 sensor, successively, were 13.2%, 22.9%, 49.0, 61.1%, 95.5%. For the SnO_2_/ZnO-2 sensor, the response values obtained were 21%, 28%, 68%, 116% and 173%, respectively, which is almost three times higher than the values obtained for the pure SnO_2_ gas sensor. In comparison to SnO_2_/ZnO-1 and SnO_2_/ZnO-2, the SnO_2_/ZnO-3 sensor possessed the best sensing performance, with a response value of 1945% to 20 ppm of NO_2_ at room temperature. Figure 5d shows the response values of the SnO_2_/ZnO-3 gas sensor to 0.1, 0.5, 5 and 10 ppm of NO_2_, which were 122%, 336%, 895% and 1384%, respectively.

The responses of SnO_2_, SnO_2_/ZnO-1, SnO_2_/ZnO-2 and SnO_2_/ZnO-3 gas sensors are shown in Figure 6a. The response of SnO_2_/ZnO was found to be better than that of pure SnO_2_ nanofibers, and it had the tendency of rising more rapidly when NO_2_ gas concentration increased. A comparison of NO_2_ gas sensing properties of different materials is shown in Table 2. From the table, it can be seen that SnO_2_/ZnO sensors exhibited high sensitivity to 0.5 ppm of NO_2_ with a response value of 336% and a faster response time of <2 min. Therefore, the response performance of SnO_2_/ZnO gas sensor was found to be remarkably higher than those of reported NO_2_ gas sensors. To further visually observe the NO_2_ sensing properties of the SnO_2_/ZnO-3 gas sensors at room temperature, the linear range for NO_2_ detection is as displayed in Figure 6b. From this figure, it can be seen that with a rise in gas concentration, there was a gradual increase in response. From Figure 6c, the response and recovery times of SnO_2_/ZnO-3 gas sensors to 0.5 ppm of NO_2_ at room temperature were 2.1 and 4.0 min, respectively. The response and recovery time of gas sensors based on as-prepared samples to 0.5 ppm of NO_2_ was as presented in Figure 6d. From this figure, it can be seen that the response/recovery time gradually decreased with the increase in the amount of ZnO doped. When compared with existing gas sensors, these results are ideal for the development of room temperature gas sensor [32].

To investigate the repeatability of SnO_2_/ZnO composites, the response performance of SnO_2_/ZnO-3 sensor to 0.3 ppm of NO_2_ was tested for four successive cycles at room temperature. As can be seen from Figure 7a, the baseline was fully capable of returning to its original position, and there was no significant difference in response values, an indication that SnO_2_/ZnO-3 gas sensor had excellent repeatability. Figure 7b shows the enlargement of response and recovery time of SnO_2_/ZnO-3 sensor to 0.3 ppm of NO_2_. It was found that the response value was 135.7%, and the response/recovery times were 1.6 and 4.0 min, respectively. Selectivity is another fundamental characteristic of gas sensors. Figure 7c shows the selectivity of SnO_2_/ZnO-3 gas sensor to 0.5 ppm of NO_2_ and 150 ppm other gases under the same measurement conditions, including HCHO, CH_4_, SO_2_, C_8_H_10_, NH_3_, and CO. The results indicate that the gas sensors based on SnO_2_/ZnO-3 had low sensitivity (response value < 5) to other gases except when compared to the values obtained for NO_2_.

When gas sensors are operated at room temperature, the effect of relative humidity on sensing properties should also be studied. To investigate the effect of humidity on both SnO_2_ sensors and SnO_2_/ZnO sensors, SnO_2_ and SnO_2_/ZnO-1 were tested in 5 ppm of NO_2_ gas under 25–96% RH at room temperature (shown in Figure 7d). Both SnO_2_ and SnO_2_/ZnO sensors worked well and had a relatively stable sensing ability, demonstrating the good humidity resistance of sensors semiconductors. Moreover, the sensing measurements were repeated every few days at room temperature to investigate the stability of SnO_2_/ZnO sensors. The results are shown in Figure 8, where the electrical signals of SnO_2_/ZnO-3 sensors did not change dramatically after 18 days when detecting NO_2_ gas. As can be seen in Figure 8d, after about ten days, the response value rose from about 250% at the beginning to about 300%, then gradually recovered and stabilized at about 250% with increasing number of days, an indication of the relatively good stability of the sensors.

### 3.3. Sensing Mechanism

During the process of NO_2_ molecule adsorption, charge transfer can occur depending on the relative band positions of SnO_2_/ZnO and NO_2_, which can cause hybridization of NO_2_ gas molecules state with SnO_2_/ZnO nanocomposite orbitals. Such a charge transfer affects the resistance of SnO_2_/ZnO, which can be facilely measured using a low-cost resistive transducing device. More importantly, physisorption of NO_2_ molecules can occur at room temperature. When compared to pure SnO_2_, SnO_2_/ZnO has a larger electronegativity that could potentially enhance its gas adsorption sites [40,41]. On the other hand, the process of chemisorption can also modulate the resistance of sensing materials. A schematic diagram of the sensing mechanism of SnO_2_/ZnO composite nanofibers to NO_2_ gas is given in Figure 9. The significant improvement in sensing properties of SnO_2_/ZnO composite nanofibers can be attributed to two factors. Firstly, the surface morphology, which is a parameter in sensing. The SnO_2_/ZnO composite nanofibers contain more SnO_2_ and ZnO nanoparticles than that of pure SnO_2_ nanofibers. This led to larger surface area (38.7 m^2^/g) in SnO_2_/ZnO composite nanofibers, providing more adsorption sites for gas molecules. Secondly, the n-n heterojunction formed at the interface of SnO_2_/ZnO composite nanofibers was also a reason for the enhanced gas sensing performance [42,43,44,45]. This phenomenon is illustrated in Figure 9a,b.

As shown in Figure 9a, the SnO_2_/ZnO composite nanofibers were composed of SnO_2_ and ZnO nanoparticles with different grain sizes. There was an energy barrier at the n–n heterojunction, which modulated the transport of electrons because of electron trapping. Figure 9b illustrates the energy band structure of SnO_2_ and ZnO, in which *E*_f_ represents the Fermi level, *E*_g_ represents the energy band gap, and *Φ*, *χ* are working function and affinity, respectively. The resistance of SnO_2_/ZnO gas sensor can be described using Equation (2):(2)$$R=R_{0}exp\left(\frac{\Delta \Phi }{k_{b}T}\right)$$
where *R*_0_ is a constant, *k_b_* is the Boltzmann’s constant, *T* is the absolute temperature, and ΔΦ is the effective potential barrier, including heterojunction and homojunction barrier [30].

In this context, oxygen molecules extracted electrons from the surface of SnO_2_/ZnO nanofibers and formed oxygen ions (like O^−^, O^−^_2_, O^2−^) in the air, thereby leading to the formation of an electron depletion layer on SnO_2_ and ZnO nanoparticles. The exact equations can be described using Equations (3)–(5):(3)$$O_{2}{+e}^{-}\to O_{2}^{-}$$
(4)$$O_{2}^{-}{+e}^{-}\to {2O}^{-}$$
(5)$$O^{-}{+e}^{-}\to O^{2-}$$

When NO_2_ gas molecules were present, they extracted electrons from SnO_2_, ZnO nanoparticles and oxygen ions because of stronger affinity to SnO_2_/ZnO. This process widened the depletion layer and increased the resistance of the gas sensor. The surface electrochemical reaction was described using Equations (6)–(9):(6)$${\mathrm{NO}}_{2}+e^{-}\to {\mathrm{NO}}_{2}^{-}$$
(7)$${\mathrm{NO}}_{2}+e^{-}\to \mathrm{NO}+O^{-}$$
(8)$${\mathrm{NO}}_{2}^{-}+O^{-}+2e^{-}\to \mathrm{NO}+2O^{2-}$$
(9)$$2{\mathrm{NO}}_{2}+O_{2}^{-}+e^{-}\to 2\mathrm{NO}+2O_{2}^{-}$$

Therefore, it can be concluded that physisorption and chemisorption play an important role in controlling the gas-sensing performance of SnO_2_/ZnO.

### 3.4. Integration and Gas Sensing Properties of Flexible Gas Sensors

The flexible electrodes were fabricated onto different flexible substrates integrated with SnO_2_/ZnO composite nanofibers to form flexible wearable gas sensors. Figure 10a shows the shapes of the electrodes prepared by screen printing with silver paste on paper. The linear silver electrodes were bent to angles of 45°, 90° and 180° and then recovered (Figure 10b) during testing for adhesion between the silver lines and paper. The results obtained showed that the adhesion was strong enough to sustain bending at any angle. Furthermore, as seen in Figure 10c, the resistance of linear screen-printed electrodes only changed from 2.0 to 2.8 Ω after bending 10,000 times to an angle of 45°. Figure 11a–c show the photographs of flexible silver electrodes for gas sensor prepared by screen printing on PDMS, paper, and PET. The silver electrode was found to have closely bonded to the substrate. The edge of the line was also clear, proving that this study had successfully prepared flexible sensors.

Figure 11d depicts the response and recovery curves of SnO_2_/ZnO-3 gas sensors with flexible silver electrodes to 0.1 ppm of NO_2_ before and after bending for 5000 and 10,000 cycles. The response values were 56%, 43% and 26%, respectively. Since the precision of screen printing is lower than that of the lithographic technique, the spacing of the interfingered electrodes was slightly larger, thereby increasing the difficulty of drop coating process. However, the idea of preparing flexible electrodes using screen printing could be extended to many other fields, such as EMI shielding [46], solar cells [47], and permanent memory devices [48]. The precision problems of screen printing can be improved through various methods such as filtration-assisted deposition [49].

## 4. Conclusions

Hollow SnO_2_ nanofibers and SnO_2_/ZnO composite nanofibers were successfully prepared through electrospinning and calcination in this work. When compared to pure SnO_2_, gas sensors based on SnO_2_/ZnO have higher sensitivity and selectivity to 0.5 ppm of NO_2_ at room temperature, with a response value of 336%. This can be attributed to the heterojunction effect and the selective NO_2_ physisorption sensing mechanism of SnO_2_/ZnO nanocomposites. In addition, the increase of the specific surface area of SnO_2_/ZnO-3 compared with pure SnO_2_ also had a positive impact on the detection sensitivity. The response and recovery time of the SnO_2_/ZnO sensors were two times shorter than those of pristine SnO_2_ sensors. In addition, flexible electrodes were fabricated using screen printing and integrated with SnO_2_/ZnO into a flexible gas sensor, then tested after 10,000 bending cycles. The flexible SnO_2_/ZnO gas sensor was able to detect 0.1 ppm of NO_2_ with a high response value of 56% at room temperature. This therefore shows that the fabrication strategy employed in this study is suitable for the development of flexible wearable sensing devices.

## Figures and Tables

**Figure 1 molecules-26-06475-f001:**
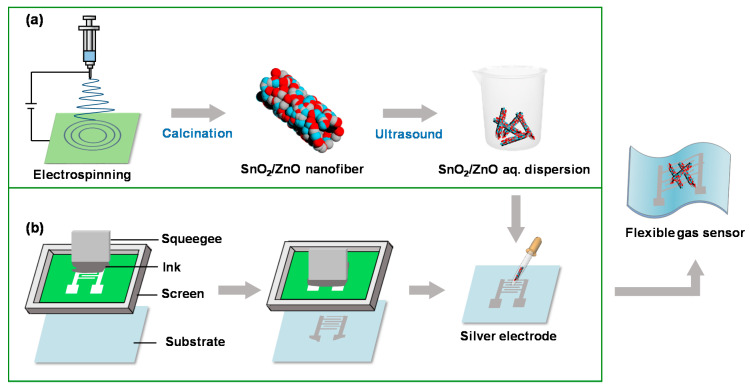
(**a**) Schematic diagrams of the preparation process of SnO_2_/ZnO composite nanofibers by electrospinning. (**b**) Fabrication process of flexible electrodes by screen printing with silver paste on substrates.

**Figure 2 molecules-26-06475-f002:**
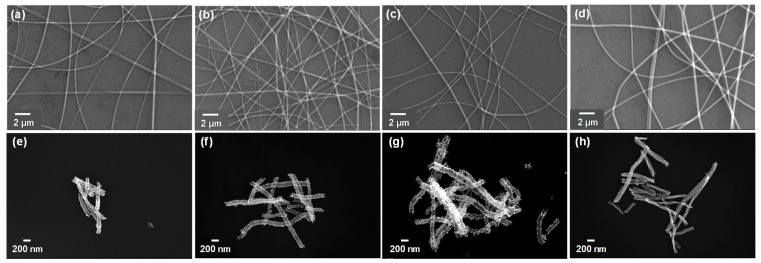
SEM images of (**a**) pristine SnO_2_/PVP, (**b**) SnO_2_/ZnO/PVP-1, (**c**) SnO_2_/ZnO/PVP-2, (**d**) SnO_2_/ZnO/PVP-3 before calcination. (**e**–**h**) Corresponding SEM images of porous hollow SnO_2_, SnO_2_/ZnO-1, SnO_2_/ZnO-2, SnO_2_/ZnO-3 nanofibers after calcination, respectively.

**Figure 3 molecules-26-06475-f003:**
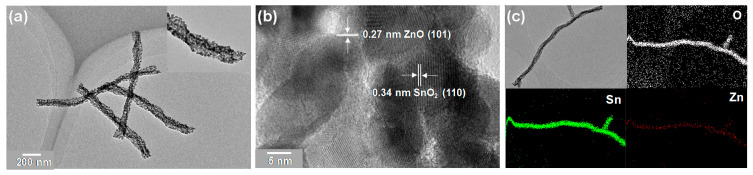
(**a**,**b**) TEM images of SnO_2_/ZnO-3 composite nanofibers. The inset of (**a**) is the enlargement. (**c**) Elemental mappings of SnO_2_/ZnO-3 composite nanofibers.

**Figure 4 molecules-26-06475-f004:**
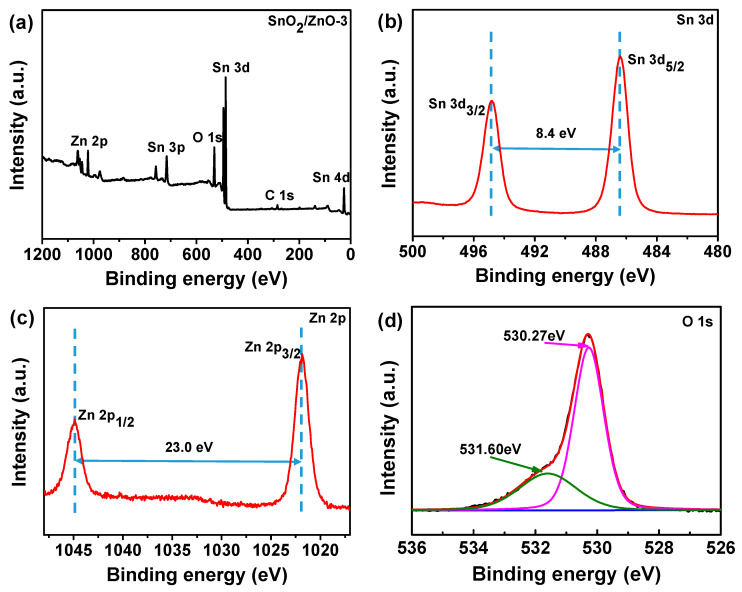
(**a**) Survey, (**b**) Sn 3d, (**c**) Zn 2p, (**d**) O 1s XPS spectra of SnO_2_/ZnO-3 composite nanofibers.

**Figure 5 molecules-26-06475-f005:**
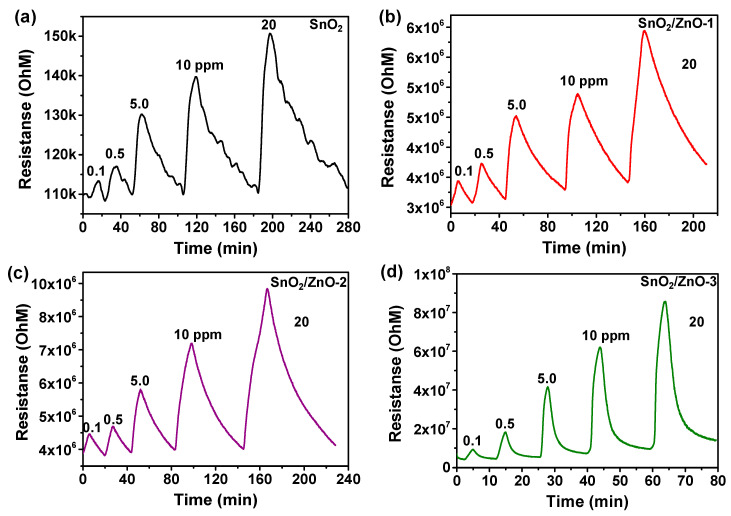
Dynamic responses of (**a**) pristine SnO_2_, (**b**) SnO_2_/ZnO-1, (**c**) SnO_2_/ZnO-2, (**d**) SnO_2_/ZnO-3 nanofibers to different concentrations of NO_2_ gas (0.1, 0.5, 5, 10, 20 ppm) at room temperature.

**Figure 6 molecules-26-06475-f006:**
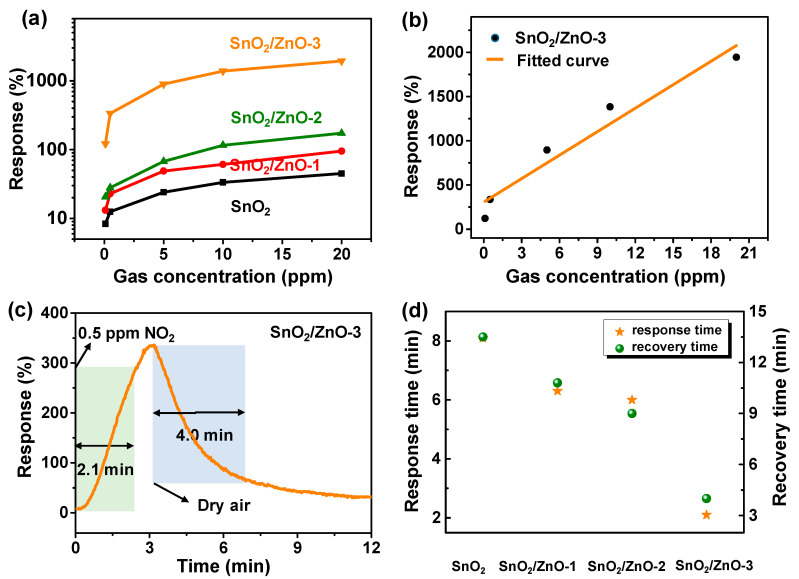
(**a**) Dynamic response curves to 0.1–20 ppm NO_2_ gas at room temperature. (**b**) The linear range for NO_2_ detection of SnO_2_/ZnO-3 gas sensors. (**c**) Response and recovery time of SnO_2_/ZnO-3 gas sensors to 0.5 ppm NO_2_. (**d**) Response and recovery time of gas sensors based on as-prepared samples to 0.5 ppm NO_2_.

**Figure 7 molecules-26-06475-f007:**
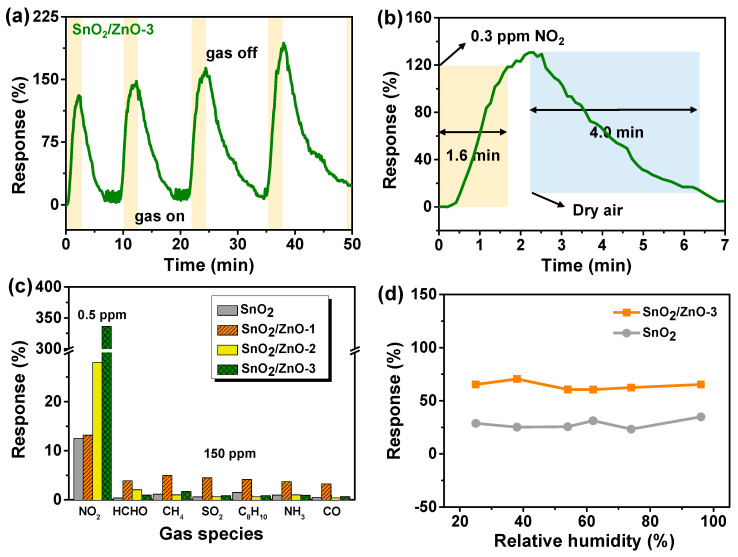
(**a**) Dynamic response of SnO_2_/ZnO-3 gas sensors to 0.3 ppm NO_2_ for four successive cycles. (**b**) Response and recovery time of SnO_2_/ZnO-3 gas sensors to 0.3 ppm NO_2_. (**c**) Selectivity of as-prepared samples sensors to 0.5 ppm NO_2_ and 150 ppm other gases (HCHO, CH_4_, SO_2_, C_8_H_10_, NH_3_, CO) under the same measurement condition. (**d**) Response vs. relative humidity of SnO_2_, SnO_2_/ZnO-3 sensors in 0.3 ppm NO_2_.

**Figure 8 molecules-26-06475-f008:**
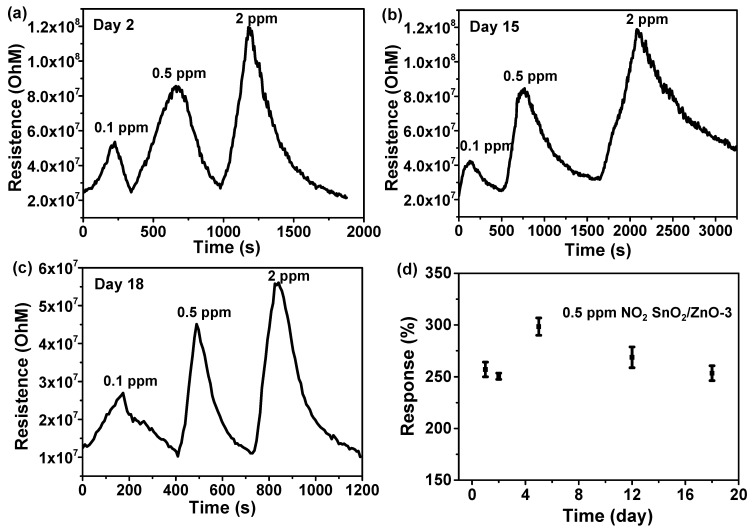
Resistance value of SnO_2_/ZnO-3 sensors to 0.1, 0.5, 2 ppm NO_2_ gas on the (**a**) second day, (**b**) 15th day, (**c**) 18th day. (**d**) The stability of SnO_2_/ZnO-3 sensors to 0.1, 0.5, 2 ppm NO_2_ gas within about 18 day.

**Figure 9 molecules-26-06475-f009:**
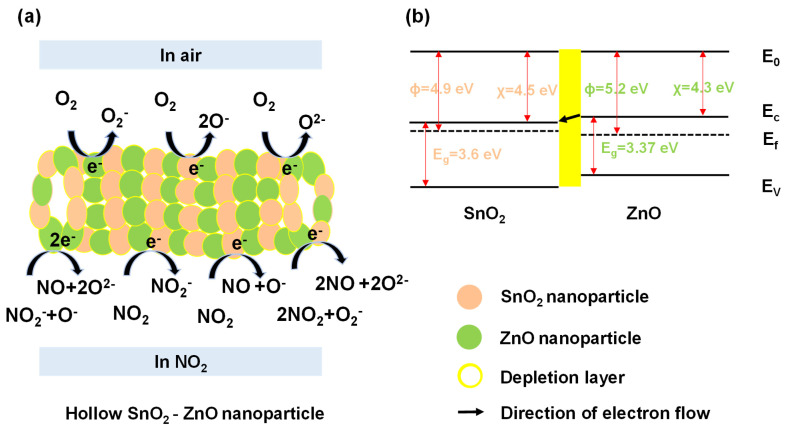
(**a**) Surface topography model of SnO_2_/ZnO composite nanofibers. (**b**) Energy band structure of SnO_2_/ZnO composite nanofibers.

**Figure 10 molecules-26-06475-f010:**
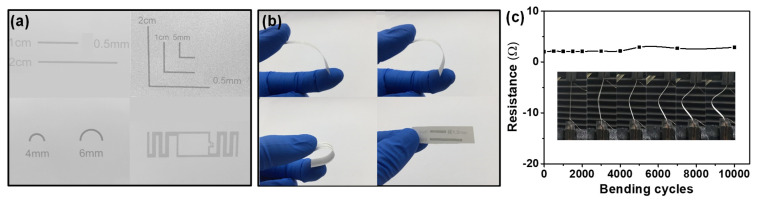
(**a**) Different shapes of electrodes prepared by screen printing with silver paste on paper. (**b**) Photographs of linear screen-printed silver electrodes bended at angles of 45°, 90°, 180° and recovered. (**c**) Resistance changes of linear screen-printed silver electrodes at angle of 45° bended for 10,000 times.

**Figure 11 molecules-26-06475-f011:**
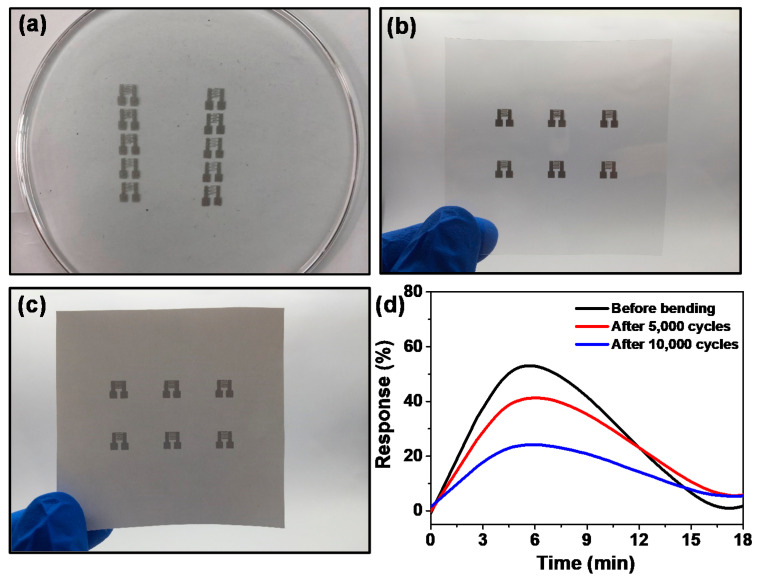
(**a**) Photograph of silver electrodes on PDMS. (**b**) Photograph of silver electrodes on paper. (**c**) Photograph of silver electrodes on PET. (**d**) Response and recovery curves of SnO_2_/ZnO-3 gas sensors on flexible silver electrodes before and after 5000, 10,000 bending cycles in 0.1 ppm NO_2_.

**Table 1 molecules-26-06475-t001:** The content of Sn, Zn, O, and C elements, and the specific surface area of SnO_2_, SnO_2_/ZnO-1, SnO_2_/ZnO-2, and SnO_2_/ZnO-3.

Samples	Sn (at.%)	Zn (at.%)	O (at.%)	C (at.%)	Specific Surface Area (m^2^/g)
SnO_2_	18.6	-	51.8	29.6	28.3
SnO_2_/ZnO-1	23.9	1.3	58.4	16.4	31.2
SnO_2_/ZnO-2	23.9	3.5	59.1	13.5	35.8
SnO_2_/ZnO-3	22.9	8.2	55.6	13.3	38.7

**Table 2 molecules-26-06475-t002:** Comparison of the NO_2_ sensing properties of different materials.

Materials	Method	°C	Response (%)	Concentration (ppm)	Response Time	Ref.
Au-WO_3_	modified precipitation/impregnation	250	836.6	5	64.2 s	[2]
Black Phosphorus	chemical exfoliation	RT	80	1	200 s	[33]
rGO-NiO	hydrothermal method	RT	100	15	300 s	[34]
MoS_2_/Graphene	annealing process	100	12.5	0.5	10 min	[35]
rGO-ZnO	solution synthesis	RT	119	1	2.4 min	[36]
Sn-doped ZnO	successive ionic layer adsorption	150	10.5	1.5	20 min	[37]
SnO_2_/ZnO	electrospin	RT	336.15	0.5	126 s	This work
SnO_2_/ZnO	thermal evaporation	RT	239	1	-	[21]
SnO_2_/ZnO	two–step hydrothermal	150	0.2	5 ppb	60 s	[30]
SnO_2_/rGO	hydrothermal treatment	50	3.31	5	135 s	[6]
rGO-Cu_2_O	nonclassic crystallization	RT	67.8	2	1000 s	[38]
rGO-Co_3_O_4_	hydrothermal method	RT	80	60	2 min	[39]

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
