# Peer review of "Highly Sensitive, Selective, Flexible and Scalable Room-Temperature NO2 Gas Sensor Based on Hollow SnO2/ZnO Nanofibers"

_molecules, 2021, doi:10.3390/molecules26216475_

Round 1

Reviewer 1 Report

Nitrogen Dioxide is one of a group of highly reactive gases known as oxides of nitrogen (NOx). NO2 interacts with water, oxygen and other chemicals in the atmosphere to form acid rain. Breathing air with some concentration of NO2 can irritate airways in the human respiratory system. It may induce various illnesses in human even at very low concentrations. Therefore, the development of sensors for the detection of nitrogen dioxide is reasonable and actual. When setting the goal of the work, the authors relied on two facts: the first was that the addition of zinc oxide to tin dioxide improves the analytical signal, and the second was the use of nanoscale materials, in particular nanofibers obtained by electrospinning, with a highly developed and active surface after calcination. This combination makes up novelty of the article and the authors managed to obtain low detection limits for nitrogen dioxide in a short time and already at room temperature. In addition, flexible electrodes were fabricated using screen printing and integrated with SnO2/ZnO into a flexible gas sensor, then tested after 10,000 bending cycles. As expected when developing a new method, the authors in Table 2 compared the parameters of the proposed sensor with those previously published in the literature. Another advantage of the article is that the authors tried to propose a mechanism for the operation of the proposed sensor.

I believe that the article can be published in this journal.

Nevertheless, I have several questions for the authors.

  1. How does heating a nanofiber to 600 degrees Celsius affect its original structure compared to that obtained by electrospinning?
  2. The authors state that the optimization of the ratio of tin dioxide and zinc oxide and the use of nanoscale fibers made it possible to obtain a high surface area of 38.7 m2 / g, which, together with the effect of heterojunction, is the reason for the high detection sensitivity of nitrogen dioxide. However, this specific surface area is not high for nanosystems and it does not depend very much on the zinc oxide content. In addition, this value should be compared with the area of pure tin dioxide.
  3. What interactions do oxides hold in nanofibers after calcination?
  4. When assessing the repeatability of the determination of nitrogen dioxide, the authors gave a picture with analytical signals, but did not give the numerical values of repeatability in percent as required by the analysis method.
  5. When assessing the selectivity of determining the nitrogen dioxide, the authors did not provide information on the effect of other nitrogen oxides, as well as triethylamine and other gases cited in the reference list.
  6. Why is there no comparison in Table 2 with the data of works 2, 6, 8, 20, 30?
  7. In reference 29 there is a mistake in the word “canofibers”.
  8. Reference 26 repeats the reference 21.

The paper can be published after minor revision.

Reviewer 2 Report

In the manuscript molecules-1351788, the authors have prepared SnO2/ZnO composite through electrospinning and calcination and demonstrated NO2 sensing at RT conditions. The manuscript is interesting and can be accepted after addressing the following concerns:

  1. While seeing the sensing results of varying compositions of SnO2-ZnO, I am concerned about SnO2/ZnO-4 or may be SnO2-ZnO-5. Have the authors checked the responses beyond SnO2/ZnO-3.?
  2. Which characterization technique was used to claim high specific surface area (38.7 m2/g) of the nanocomposite. I can’t find any BET results here.
  3. Also, have the authors checked the sensing responses at varied temperatures. Slightly varying the operating temperature might bring down the response and recovery times. Sensing responses should be checked upto 100 C.
  4. Why SnO2/ZnO-1 was chosen for results in Figure 7d, when the SnO2/ZNO-3 showed the best results.
  5. Figure 6a: Is the sensing response saturated after 20 ppm. Please provide the results beyond 20 ppm upto 100ppm or upto 500 ppm.
  6. Figure 7c: it is not justifiable to compare the NO2 with the reducing gases (CH4, HCHO, C8H10) at room temperature conditions.
  7. I don’t see the stability as ‘good’ in figure 8d as claimed by authors. Please include standard deviation and error bars for these results.
  8. Important reference for SnO2/ZnO sensor can be cited: https://doi.org/10.1063/1.5123479
